# An Overview of the Safety Profile and Clinical Impact of CDK4/6 Inhibitors in Breast Cancer—A Systematic Review of Randomized Phase II and III Clinical Trials

**DOI:** 10.3390/biom13091422

**Published:** 2023-09-20

**Authors:** Ioana-Miruna Stanciu, Andreea Ioana Parosanu, Cornelia Nitipir

**Affiliations:** 1Department of Oncology, “Carol Davila” University of Medicine and Pharmacy, 020021 Bucharest, Romania; ioana-miruna.stanciu@drd.umfcd.ro (I.-M.S.);; 2Department of Medical Oncology, Elias University Emergency Hospital, 011461 Bucharest, Romania

**Keywords:** breast cancer, CDK4/6 inhibitor toxicities, safety profile, palbociclib, ribociclib, abemaciclib

## Abstract

Cyclin-dependent kinase 4 and 6 inhibitors (CDK4/6is) have transformed the treatment of hormone receptor-positive (HR+) and human epidermal growth factor receptor 2-negative (HER2−) breast cancer over the last decade. These inhibitors are currently established as first- and second-line systemic treatment choices for both endocrine-sensitive and -resistant breast cancer populations alongside endocrine therapy (ET) or monotherapy. Data on targeted therapy continue to mature, and the number of publications has been constantly rising. Although these drugs have been demonstrated to prolong overall survival (as well as progression-free survival (PFS) in breast cancer patients), changing the paradigm of all current knowledge, they also cause important adverse events (AEs). This review provides the latest summary and update on the safety profile of the three CDK4/6 inhibitors, as it appears from all major phase II and III randomized clinical trials regarding palbociclib, ribociclib, and abemaciclib, including the most relevant 15 clinical trials.

## 1. Introduction

Breast cancer (BC) is the second leading cause of cancer death in women after lung cancer, and the chance that a woman will die from breast cancer is about 1 in 39 (about 2.5%), according to statistics from the American Cancer Society [1]. It is the most occurring cancer in women and the most common cancer overall, with more than 2.26 million new cases diagnosed in 2020 [2]. It has become an increasingly severe disease burden threatening women’s health [3]. By the end of 2023, an estimated 300,590 people (297,790 women and 2800 men) in the United States (US) will be diagnosed with invasive BC. In the US, 6% of women have metastatic BC when they are first diagnosed, with a 5-year relative survival rate of 30% [4].

Around 75% of BC patients are diagnosed with a hormone receptor-positive (HR+), human epidermal growth factor receptor 2-negative (HER2−) type of BC [5]. The current treatment for HR+/HER2− metastatic BC involves endocrine therapy (ET), targeted therapy, and chemotherapy. The game changers of this type of BC are three cyclin-dependent kinase 4 and 6 inhibitors (CDK4/6is) already approved by the US Food and Drug Administration (palbociclib, ribociclib, and abemaciclib) in combination with ET for first-line therapy [6,7,8]. Although they all possess the same mechanism of action, these drugs have slight differences in efficacy and safety.

The combination of a CDK4/6i and an AI is efficacious even for patients with visceral disease. Approximately 45% to 60% of patients enrolled across all first-line studies had visceral disease. In subgroup analyses, these patients benefited similarly to the overall study populations. Given its robust overall response rate (50% to 60% in the first line), the CDK4/6i and AI may be considered even when rapid tumor response is needed, although chemotherapy remains a gold standard in case of visceral crisis [9,10].

The CDK4/6is are generally well tolerated, but, like any other drug, they still have AEs. The most common adverse effects are nausea, fatigue, diarrhea, neutropenia, anemia, leukopenia, and thrombocytopenia [11]. However, the neutropenia associated with CDK4/6 inhibitors is distinct from chemotherapy-induced neutropenia in that it is rapidly reversible, reflecting a cytostatic effect on neutrophil precursors in the bone marrow [12]. Most hematologic abnormalities seen with CDK4/6 inhibitors are uncomplicated and adequately managed with standard supportive care and dose adjustments when indicated. Palbociclib and ribociclib most commonly cause neutropenia, while diarrhea is the most common adverse effect of abemaciclib, perhaps because of its greater affinity for CDK4 over CDK6 [10,13].

As multiple research articles have been published on this topic in the literature over the last few years, this review aims to crystallize the current relevant results. In the next paragraphs, our attention is focused on the mechanism of action of CDK4/6is, the development of resistance to CDK4/6is over time, the safety profile of CDK4/6is, and the newest data regarding clinical outcomes.

## 2. Aims and Objectives

The greatest challenge in the research and development of CDK4/6 inhibitors might lie in dealing with the adverse effects and potential drug tolerance. Further understanding of the underlying mechanism and exploring the ideal combination therapy might help to overcome the selectivity and drug tolerance of CDK inhibitors. Considering the similar efficacies and indications of palbociclib, ribociclib, and abemaciclib, the evaluation of their toxicity profiles may facilitate treatment choice. This review explores the safety profiles of palbociclib, ribociclib, and abemaciclib in all randomized phase II and III clinical trials.

## 3. Materials and Methods

This systematic review was in accordance with Preferred Reporting Items for Systematic Reviews and Meta-Analyses guidelines [14]. Extended systematic research was performed on ClinicalTrials.gov database clinical trial registries with results published or registered before July 2023. The keywords used were palbociclib, ribociclib, abemaciclib, phase II and III, randomized, and available results. The last search was performed on the 8th of August 2023. The search returned 690 results.

Literature articles were systematically assessed by searching in the PubMed Medline database from January 2016 until July 2023. The following keywords were used: CDK4/6 inhibitors, BC, and toxicities. The inclusion criteria were clinical studies on BC patients, clinical phase II and III studies using currently approved doses for palbociclib (125 mg/day orally for 3 weeks, followed by 1 week off), ribociclib (200 mg × 3/day orally for 3 weeks, followed by 1 week off) or abemaciclib (150 mg × 2/day orally in combination with ET or 200 mg/day orally in monotherapy) and studies for which safety data of at least one AE were reported in the article in the form of the percentage or number of patients. Only free full-text articles were assessed. The exclusion criteria were articles with unavailable abstracts, non-English articles, conference presentations, meta-analyses and literature reviews, non-clinical studies, post hoc or subgroup analyses, retrospective studies, and studies in which adverse effects were not reported. In vivo and in vitro studies were also excluded. The search returned 130 results, all of which were already included in the first search on ClinicalTrials.gov; therefore, it was chosen to present below the selection of clinical trials from the ClincialTrials.gov database (Figure 1).

### The Rationale of This Review

This review is intended to assist healthcare professionals who wish to deepen their knowledge of the toxicities associated with CDK 4/6i therapy. The complexity of the adverse reactions encountered raises problems for both patients and medical personnel, and their rapid and adequate management can prevent the occurrence of further complications. From the experience of our clinic, a majority of patients under treatment with CDK4/6is face at least one adverse reaction during treatment. Therefore, it is of major importance to acknowledge every one of them and treat it accordingly. Moreover, not only results from clinical trials but also real-world data demonstrated that there is a significant improvement in OS and PFS, which makes us full of hope regarding future perspectives in treating breast cancer patients.

## 4. Review

### 4.1. The Efficacy of CDK4/6is

Usage of CDK4/6 inhibitors has led to improvement in PFS and OS in patients with HR+/HER2− advanced breast cancer.

Patients treated with CDK4/6 inhibitors combined with ET had significantly longer PFS and significantly better overall response rates (ORR) and clinical benefit rates (CBR) compared to those treated with placebo combined with ET, according to Li et al.’s systematic review and meta-analysis of the efficacy and safety of CDK4/6 inhibitors from 2021. This finding suggested that patients with advanced BC and HR+/HER-2 could dramatically improve their short-term prognosis by combining CDK4/6 inhibitors and ET. Further examination of OS in the PALOMA-3, MONALEESA-3, MONALEESA-7, and MONARCH-2 studies further demonstrated that, in comparison to those receiving endocrine monotherapy, HR+/HER-2 advanced BC patients receiving CDK4/6 inhibitors in combination with ET also experienced considerably longer OS. For patients with a heavy tumor burden, such as bone metastasis and visceral metastasis, it was found that CDK4/6 inhibitors plus ET can still achieve satisfactory therapeutic effects [15].

A single institutional study from India, conducted by Ganguly and colleagues on 144 patients treated with CDK4/6is, showed that the median PFS of the whole population was 16 months, and OS was 29.1 months [16].

A very recent systematic review and meta-analysis conducted by Potrelli et al. demonstrated that elderly patients (those aged 65 years) with advanced ER+ BC have good OS and PFS from CDK4/6 inhibitors [17]. This suggests that after geriatric assessment and in accordance with the toxicity profile, the CDK4/6i therapy could be safely prescribed even to elderly patients—not only to those under 65 years of age. However, no subgroup analysis has been conducted on this population.

### 4.2. The Rationale for CDK4/6 Inhibitors and Their Limitations

Given the striking success of the combination of CDK4/6 inhibitors and ET in BC, many other combinations have recently entered clinical trials in multiple diseases. To maximally benefit from CDK4/6 inhibitors, understanding how CDK4/6 inhibitors work will be critical [18]. In the next paragraphs, the mechanisms by which CDK4/6 inhibitors can exert their anti-tumor activities beyond simply enforcing cytostatic growth arrest are highlighted, and the primary and achieved resistance to CDK4/6is is discussed.

In normal conditions, the cell cycle transitions through several distinct phases: G0 (quiescence) followed by G1 (pre-DNA synthesis), S (DNA synthesis), G2 (pre-division), and M (cell division). Dysregulation of one or more of these results in abnormal replication, chaotic division, and, finally, cancer [19].

CDK4/6 specifically regulates the cellular transition from the G1 phase of the cell cycle to the S phase. Therefore, the CDK4/6 inhibitors effectively block the proliferation of sensitive cancer cells by arresting the G1 cell cycle [19]. The inhibition occurs by blocking the retinoblastoma protein (pRB) phosphorylation, which inactivates pRB and releases transcription factors that allow progression to the S phase [20].

Recent studies have shown that, besides blocking the cell cycle, CDK4/6 inhibitors also suppress tumor growth through multiple other mechanisms, including enhancing the cytostasis caused by signaling-pathway inhibitors, inducing senescence, regulating cell metabolism, and even promoting anti-tumor immune responses [18,21].

However, approximately 10% of patients will have primary resistance to CDK4/6 inhibitors [22]. The possible mechanisms of resistance to CDK4/6 inhibitors and predictive biomarkers of response were assessed in a review of the literature performed from January 2013 to January 2023 [23]. For instance, patients with evidence of functional pRB loss at baseline or with increased cyclin E1/E2 expression are not likely to benefit from CDK4/6 inhibition [23]. A rise in thymidine kinase 1 activity may also provide a marker of early resistance. Mutations in pRB resulting in the activation of other cell cycle factors, such as E2F and the Cyclin E-CDK2 axis, as well as PI3K/AKT/mTOR pathway activation, have been demonstrated in the cases of acquired resistance to CDK4/6 inhibitors + ET [23].

### 4.3. Safety Profile of Palbociclib

PALOMA-1 is a phase II, open-label, randomized study of letrozole plus palbociclib versus letrozole as a single agent for the first-line treatment of HR+/HER2− advanced BC in postmenopausal women [24]. Neutropenia, leucopenia, and weariness were the most prevalent AE observed in the palbociclib plus letrozole group. There was at least one AE in all 83 individuals receiving palbociclib with letrozole, compared to 65 (84%) of 77 patients who received letrozole alone. Despite the fact that palbociclib with letrozole increased all grades of neutropenia and leucopenia, no incidences of neutropenic fever were documented. Anemia, nausea, arthralgia, and alopecia were among the other AEs (of any source) that were more common in the palbociclib plus letrozole group, but the majority of these were minor. The following serious AEs occurred in more than one patient in the palbociclib plus letrozole group: pulmonary embolism (4% of patients), back pain (2% of patients), and diarrhea (2% of patients). In the letrozole group, no significant AEs occurred in more than one patient. There were 33% of patients in the palbociclib + letrozole group with dosage interruptions from AEs, compared to only three (4%) individuals in the letrozole group. Because of an adverse event, 37 (45%) patients in the combination group required a delay in the start of a subsequent treatment cycle, and 33 (40%) required a dosage decrease. Disease progression was the primary reason for trial withdrawal in both therapy groups (42 (50%) patients in the palbociclib + letrozole group versus 57 (70%) individuals in the letrozole group). Because of an adverse event, 11 (13%) patients in the palbociclib + letrozole group and 2 (2%) in the letrozole group dropped out of the study. Six (7%) participants in the palbociclib + letrozole group and two (2%) in the letrozole group stopped due to treatment-related side events. During the research, one fatality occurred in the palbociclib plus letrozole group due to illness progression; no treatment-related fatalities occurred [25].

The PALOMA-2 trial is a randomized, multicenter, double-blind, phase III study comparing the clinical benefit following treatment with letrozole in combination with palbociclib versus letrozole in combination with placebo in postmenopausal women with ER+/HER2− advanced BC who have not received prior systemic anti-cancer therapies for their advanced/metastatic disease [26]. The findings presented here reflect palbociclib exposure in 444 of 666 patients with advanced HR+/HER2− BC who received at least one dose of palbociclib with letrozole. The median length of palbociclib plus letrozole treatment was 19.8 months, while that of placebo plus letrozole was 13.8 months. Of patients receiving a placebo with letrozole, 36% had their dose reduced due to an adverse response of any severity. Permanent cessation due to an adverse response occurred in 43 of 444 (9.7%) palbociclib plus letrozole patients and 13 of 222 (5.9%) placebo plus letrozole patients. Neutropenia (1.1%) and an increase in alanine aminotransferase (0.7%) were among the adverse responses resulting in permanent cessation in individuals receiving a placebo with letrozole. In descending frequency, the most common adverse reactions of any grade reported in patients in the palbociclib plus letrozole arm were neutropenia, infections, leukopenia, fatigue, nausea, alopecia, stomatitis, diarrhea, anemia, rash, asthenia, thrombocytopenia, vomiting, decreased appetite, dry skin, and dysgeusia (all 10%). In patients taking palbociclib with letrozole, the most often reported grade 3 AEs (5%), in decreasing order, were neutropenia, leukopenia, infections, and anemia [27,28].

The PALOMA-3 study is a randomized, double-blind, placebo-controlled, phase III clinical trial with the primary objective of demonstrating the superiority of palbociclib in combination with fulvestrant over fulvestrant alone in prolonging PFS in women with HR+/HER2− metastatic BC whose disease has progressed after prior ET [29]. The findings presented here indicate palbociclib exposure in 345 of 517 patients with advanced or metastatic HR+/HER2− BC who received at least one dose of palbociclib with fulvestrant. The median length of palbociclib plus fulvestrant treatment was 10.8 months, while the median duration of placebo plus fulvestrant treatment was 4.8 months. Neutropenia of grade 3 or 4 occurred in 70% of patients receiving palbociclib-fulvestrant but not in any of them receiving placebo-fulvestrant, whereas anemia of grade 3 or 4 occurred in 4% and 2% of patients, respectively, and thrombocytopenia of grade 3 or 4 occurred in 3% and none of the patients, respectively [6]. Infections (in 5% of the palbociclib-fulvestrant group and 3% of the placebo-fulvestrant group), fatigue (in 3% and 1%, respectively), and elevation in the aspartate aminotransferase level (in 3% and 2%, respectively) occurred at a frequency of more than 2% of the patients in the palbociclib-fulvestrant group [6]. Of patients taking palbociclib + fulvestrant, 36% had their dose reduced due to an adverse event of any severity. Permanent cessation due to an adverse response occurred in 19 of 345 (6%) palbociclib plus fulvestrant patients and 6 of 172 (3%) placebo plus fulvestrant patients. Patients taking palbociclib with fulvestrant had tiredness (0.6%), infections (0.6%), and thrombocytopenia (0.6%), which led to cessation. In descending frequency, the most frequent adverse effects of any grade observed in patients in the palbociclib plus fulvestrant arm were neutropenia, leukopenia, infections, tiredness, nausea, anemia, stomatitis, diarrhea, thrombocytopenia, vomiting, alopecia, rash, and reduced appetite (all 10%). The most frequently reported grade ≥3 adverse reactions (≥5%) in patients receiving palbociclib plus fulvestrant in descending frequency were neutropenia and leukopenia [28].

PALOMA-4 is a randomized, double-blind phase III study of palbociclib plus letrozole versus placebo plus letrozole for the treatment of previously untreated Asian postmenopausal women with HR+/HER2− advanced BC [30]. The most common all-grade AEs in the palbociclib plus letrozole arm were neutropenia (84.5% vs. 1.2%), leukopenia (36.3% vs. 0.6%), thrombocytopenia (6.5% vs. 0.6%), and anemia (4.8% vs. 1.8%). Four patients in the palbociclib arm developed febrile neutropenia. Any SAE occurred in 15.5% of individuals in the palbociclib arm and 9.4% of participants in the placebo arm; infections were the most prevalent SAE (2.4% vs. 2.9%). Ten people died (five on palbociclib and five from the placebo group). One fatality in the palbociclib arm was thought to be attributable to the study medication [31].

PALLAS is a randomized, open-label phase III study evaluating the addition of 2 years of palbociclib to standard adjuvant ET for patients with HR+/HER2− early BC [32]. The safety population included 5743 patients: 2840 on palbociclib + endocrine treatment and 2903 on ET alone. Treatment-emergent AEs occurred in 2822 (99.4%) of the 2840 patients who received palbociclib + endocrine treatment and in 2571 (88.6%) of the 2903 who received ET alone. Neutropenia (1742 (61.3%) of 2840 patients on palbociclib with endocrine treatment versus 11 (0.3%) of 2903 on ET alone), leucopenia (857 (30.2%) vs. 3 (0.1%)), and tiredness (60 (2.1%) vs. 10 (0.3%)) were the most prevalent grade 3–4 side effects. Febrile neutropenia was infrequent (28 (1.0%) patients receiving palbociclib plus endocrine treatment vs. none (0%) receiving ET alone). The incidence of grade 3–4 neutropenia in patients receiving palbociclib with endocrine treatment was 63.1% (1487 of 2355) in those who had previously had chemotherapy, against 52.6% (255 of 485) in those who had not. By 6 months, 42.2% of patients needed at least one dose decrease of palbociclib to 100 mg, 48.9% at 12 months, 53.5% at 18, and 55.4% at 24. Interstitial lung disease occurred in 15 of 2840 patients on palbociclib + ET vs. 5 of 2903 patients on ET alone, according to a post hoc review of specific side events. Venous thromboembolism and concomitant thrombotic AEs occurred in 47 individuals vs. 20. SAEs occurred in 351 of 2840 palbociclib + endocrine treatment patients vs. 220 of 2903 ET-alone patients. Tissue infection (49 individuals on palbociclib with endocrine treatment versus 29 on ET alone) and upper respiratory tract infection (23 vs. 3) were the most prevalent major side effects [33].

PALLET is a randomized phase II study evaluating palbociclib in addition to letrozole as neoadjuvant therapy in HR+ early BC. An adverse event of any grade was recorded in 91% of letrozole patients and 99% of palbociclib + letrozole patients. The majority of AEs (91%) were grade 1 or 2. Grade 3 or higher AEs were recorded in 17% of letrozole patients and 50% of palbociclib + letrozole patients. In all, eight patients in the palbociclib + letrozole groups had ten grade 4 or 5 AEs. One of these patients developed a grade 5 acute respiratory distress syndrome thought to be unrelated to letrozole or palbociclib [34].

We present below (Table 1) the adverse reactions (≥10%) reported in the PALOMA-1, PALOMA-2, PALOMA-3, PALOMA-4, PALLAS, and PALLET studies for the palbociclib arms.

### 4.4. Safety Profile of Ribociclib

MONALEESA-2 is a phase III randomized, double-blind, placebo-controlled study of ribociclib combined with letrozole for the treatment of postmenopausal women with HR+/HER2− advanced BC who received no prior therapy for advanced disease [35]. In the safety population (334 patients in the ribociclib group and 330 in the placebo group), AEs of any grade that occurred in at least 35% of the patients in either group were neutropenia (74.3% in the ribociclib group and 5.2% in the placebo group), nausea (51.5% and 28.5%, respectively), infections (50.3% and 42.4%), fatigue (36.5% and 30.0%), and diarrhea (35.0% and 22.1%) [36].

Neutropenia was the most prevalent grade 3 or 4 AE, occurring in 63.8% of ribociclib patients and 1.2% of placebo patients. Febrile neutropenia occurred in five patients (1.5%) in the ribociclib group and none in the placebo group. Hepatobiliary toxic effects (14.4% and 4.8%, respectively) and extended QT interval (4.5% and 2.1%, respectively) were other grade 3 or 4 AEs of particular relevance. None of these cases resulted in death, and aminotransferase and bilirubin levels returned to normal in all four patients after discontinuing ribociclib. Two patients (0.6%) in the ribociclib group and none in the placebo group had grade 3 interstitial lung disease or pneumonitis. The ribociclib group showed no grade 4 AEs or fatalities associated with interstitial lung disease or pneumonitis [37].

SAEs occurred in 71 (21.3%) of the ribociclib and 39 (11.8%) of the placebo patients. The trial regimen was assessed to be responsible for 25 (7.5%) of these events in the ribociclib and 5 (1.5%) in the placebo group. During therapy, there were four fatalities (three (0.9%) in the ribociclib and one (0.3%) in the placebo group). Each group lost one patient due to the development of the underlying BC. The other two deaths in the ribociclib group occurred as a result of sudden death and death from an unexplained cause. The abrupt death was thought to be caused by ribociclib and happened on day 11 of cycle 2 in conjunction with grade 3 hypokalemia and a grade 2 prolongation in the QT interval. The woman who died from an unknown cause was prescribed ribociclib for four days before withdrawing consent and ceasing the trial therapy; her death was reported 19 days later, and the investigator determined that it was unconnected to ribociclib [36].

MONALEESA-3 is a phase III randomized, double-blind, placebo-controlled study of ribociclib in combination with fulvestrant for treating postmenopausal women and men with HR+/HER2− advanced BC who have received no or only one line of ET for advanced BC [38]. The safety population had 724 patients. Neutropenia, nausea, and tiredness were the most prevalent of all-grade AEs observed in 30% of patients in either group. Neutropenia and leukopenia were the most prevalent grade 3 AEs, occurring in 10% of patients. Neutropenia was the only grade 4 incident observed in 5% of patients. Febrile neutropenia occurred in 1.0% of ribociclib plus fulvestrant patients versus 0% of placebo plus fulvestrant patients. ECG QT prolongation (of any grade) occurred in 6.2% of ribociclib plus fulvestrant patients and 0.8% of placebo plus fulvestrant patients. Grade 3 or 4 increased ALT occurred in 32 (6.6%) and 9 (1.9%) individuals in the ribociclib plus fulvestrant arm, respectively, whereas elevated AST occurred in 23 (4.8%) and 6 patients (1.2%), respectively. SAEs occurred in 138 (28.6%) and 40 (16.6%) participants in the ribociclib plus fulvestrant and placebo plus fulvestrant arms, respectively; of these, 54 (11.2%) and 6 (2.5%) were ascribed to the study medication. Ribociclib or placebo dosage decreases were recorded in 183 (37.9%) and 10 (4.1%) patients in the ribociclib plus fulvestrant and placebo plus fulvestrant groups, respectively. During or within 30 days after treatment discontinuation, 13 died (2.7%) in the ribociclib plus fulvestrant arm and 8 (3.3%) in the placebo plus fulvestrant arm; disease progression caused the majority of deaths. The ribociclib plus fulvestrant group had one fatality due to acute respiratory distress syndrome in a patient with baseline lung metastases, which was thought to be related to study therapy. The remaining five fatalities had nothing to do with therapy [39].

MONALEESA-7 is a phase III randomized, double-blind, placebo-controlled study of ribociclib or placebo in combination with tamoxifen and goserelin or a non-steroidal aromatase inhibitor (NSAI) and goserelin for the treatment of premenopausal women with HR+/HER2− advanced BC [40]. AEs in the two groups remained consistent with those in the primary analysis. Grade 3 or 4 AEs of special interest were neutropenia (63.5% of patients in the ribociclib group and 4.5% in the placebo group), hepatobiliary toxic effects (11% and 6.8%, respectively), and prolonged QT interval (1.8% and 1.2%, respectively) [37].

CORALLEEN is an open-label, multicenter, randomized, phase II trial of ribociclib plus letrozole versus chemotherapy for postmenopausal women with HR+/HER2− luminal B BC [41]. In the ribociclib plus letrozole group, the most common grade 1–2 AEs were neutropenia (29 (57%) of 51 patients), elevated alanine aminotransferase (13 (26%)), and asthenia (13 (26%)). In the ribociclib plus letrozole group, 29 (57%) of 51 patients and 36 (69%) of 52 individuals in the chemotherapy group experienced grade 3–4 toxicities. In the ribociclib plus letrozole group, the most prevalent grade 3–4 AEs were neutropenia (22 (43%) of 51 patients) and elevated alanine aminotransferase concentrations (10 (20%)). Grade 1–3 QTc prolongation was found in 2 (4%) of 51 patients in the ribociclib plus letrozole group, while 8 (16%) of 51 patients ended study treatment due to grade 3–4 elevated alanine and aspartate aminotransferases. Four (8%) of the fifty-two individuals required a ribociclib dosage decrease [42].

We present below (Table 2) the adverse reactions (≥10%) reported in the MONALEESA-2, MONALEESA-3, MONALEESA-7, and CORALLEEN studies for the ribociclib arms.

### 4.5. Safety Profile of Abemaciclib

MONARCH-2 is a randomized, double-blind, placebo-controlled, phase III study of fulvestrant with or without abemaciclib for women with HR+/HER2− locally advanced or metastatic BC [43]. In the included safety population, 441 received abemaciclib, and 223 received placebo. The most frequent AEs of any grade were diarrhea, neutropenia, nausea, fatigue, and abdominal pain, predominantly of grade 1 or 2 severity. In the abemaciclib arm, febrile neutropenia was noted in six patients. One of these had grade 2 afebrile neutropenia, mistakenly recorded as febrile neutropenia, while another experienced febrile neutropenia 53 days after stopping abemaciclib but had already received post-study paclitaxel. The final four cases of febrile neutropenia were unrelated to grade 3 or higher infections. In both arms, little granulocyte colony-stimulating factor was used. Regardless of relatedness, the abemaciclib arm had a greater infection rate (42.6%) than the placebo arm did (24.7%); however, the majority of these infections were grade 1 to 2 in severity (6.6% in the abemaciclib arm vs. 3.6% were grade 3). SAEs were observed in 22.4% of abemaciclib patients and 10.8% of placebo patients. SAEs likely related to the study drugs were reported in 8.8% of the abemaciclib arm and 1.3% of the placebo arm, with diarrhea being the most common (1.4% in the abemaciclib arm vs. 0% in the placebo arm). The most commonly reported SAEs were thromboembolic events, which occurred in nine patients (2.0%) in the abemaciclib arm and one (0.4%) in the placebo arm. Four patients in the abemaciclib arm developed pulmonary embolisms, none of which resulted in death. In the abemaciclib arm, 322 patients (73.0%) and 54 (24.2%) in the control arm experienced grade 1 or 2 diarrhea. Grade 3 diarrhea, on the other hand, was less common (n = 59 (13.4%) vs. n = 1 (0.4%) in the abemaciclib and control arms, respectively). Diarrhea episodes were most common in the abemaciclib arm during the first treatment cycle (median start of diarrhea was 6 days). Most episodes of diarrhea were efficiently treated with antidiarrheal medicines and dosage changes. Of patients in the abemaciclib arm, 14.5% who had an initial grade 2 diarrhea incident and 1.1% who had an initial grade 3 diarrhea event had a recurrence at the same or higher degree. The majority (70.1%) of patients who experienced diarrhea in the abemaciclib arm did not require treatment adjustment (i.e., dosage interruption, decrease, or termination); nonetheless, 2.9% of patients stopped taking the abemaciclib due to diarrhea. The most prevalent abnormalities based on central laboratory tests were elevated serum creatinine levels, reduced white blood cells and neutrophil counts, and anemia. The abemaciclib arm had 14 (3.2%) fatalities (9 due to AEs) and 10 (4.5%) in the control arm (2 due to AEs) while on medication or within 30 days of treatment termination. Three deaths (0.7%) in the abemaciclib arm were determined to be related to study treatment, two due to sepsis in patients who did not follow guidance on granulocyte colony-stimulating factor administration and dose reduction, and one due to viral pneumonia in a patient receiving steroids for spinal stenosis. [8]

MONARCH-3 is a randomized, double-blind, placebo-controlled, phase III study of NSAIs (anastrozole or letrozole) plus abemaciclib, or placebo in postmenopausal women with HR+/HER2− locoregionally recurrent or metastatic BC with no prior systemic therapy. The most common AEs reported by the investigator in the abemaciclib arm (n = 327 in the abemaciclib arm; n = 161 in the placebo arm) were diarrhea, neutropenia, exhaustion, and nausea. The most prevalent abnormalities based on central laboratory tests were increased serum creatinine, reduced white blood cell and neutrophil counts, and anemia. SAEs were recorded in 27.5% of abemaciclib patients and 14.9% of placebo patients, with lung infection being the most common (2.8% vs. 0%, respectively). Diarrhea was mostly mild (abemaciclib arm vs. placebo arm, 44.6% vs. 21.7%; grade 2: 27.2% vs. 6.8%). The median start for the abemaciclib arm was 8.0 days, and the median duration was 10.5 days (grade 2) and 8.0 days (grade 3). The majority of patients (76.3%) who experienced diarrhea in the abemaciclib arm did not have their therapy modified. Of the individuals who had diarrhea, 73.3% used antidiarrheal medication. The rate of study treatment discontinuation due to diarrhea was 2.3% in the abemaciclib arm. Of individuals in the abemaciclib arm, 41.3% observed neutropenia. Overall, once reduced, the neutrophil count remained constant during abemaciclib administration and was reversible when treatment was stopped. In the abemaciclib arm, one patient had nonserious febrile neutropenia. Infections occurred in 39.1% of patients in the abemaciclib arm and 28.6% in the placebo arm, with the majority of infections being grade 1 or 2 (33.3% in the abemaciclib arm vs. 25.5% in the placebo arm). Venous thromboembolic events occurred in 16 (4.9%) of abemaciclib participants compared to 1 (0.6%) in the placebo arm. The majority of patients in the abemaciclib arm (11 of 16) did not stop therapy (4 had dosage disruptions at the time of the occurrence). Increased ALT abnormalities were seen in 47.6% of patients in the abemaciclib arm versus 25.2% in the placebo. In the abemaciclib arm, 36.7% of patients had increased AST, compared to 23.2% in the placebo [44].

NEXT MONARCH is a randomized, open-label, phase II study of abemaciclib plus tamoxifen or abemaciclib alone in women with previously HR+/HER2− metastatic BC [45]. Dose omissions due to AEs occurred in 32 patients (41.0%) in the abemaciclib + tamoxifen arm, 38 patients (48.1%) in the abemaciclib 150 mg arm, and 50 patients (64.9%) in the abemaciclib 200 mg arm, with the most common causes being neutropenia (60.3%), diarrhea (18.0%), leukopenia (12.8%), and thrombocytopenia (11.6%). Abemaciclib dosage reductions were necessary in 21 patients (26.9%) in the abemaciclib + tamoxifen arm, 25 patients (31.6%) in the abemaciclib 150 mg arm, and 38 patients (49.4%) in the abemaciclib 200 mg arm, with the majority requiring just one decrease. Neutropenia (32.1%), diarrhea (15.5%), and leukopenia (12.9%) were the most prevalent reasons for dosage decreases. The most frequently reported treatment-emergent AEs of any grade, regardless of causality, in ≥30% of patients included diarrhea (61.1%), neutropenia (47.9%), anemia (37.6%), and nausea (35.5%). Neutropenia (28.6%), leukopenia (11.5%), anemia (9.8%), and thrombocytopenia (5.1%) were the most common grade ≥3 AEs independent of etiology, which occurred in 5% of individuals. Two fatal events were recorded in each arm related to AEs while on treatment or within 30 days of discontinuation: cardiac arrest and disseminated intravascular coagulation in the abemaciclib + tamoxifen arm [46].

MONARCH PLUS is a randomized, double-blind, placebo-controlled, phase III study to compare anastrozole or letrozole plus abemaciclib or plus placebo and to compare fulvestrant plus abemaciclib or plus placebo in postmenopausal women with HR+/HER2− locoregionally recurrent or metastatic BC [47]. Neutropenia, diarrhea, leukopenia, and anemia were the most commonly reported treatment-emergent AEs in both cohorts of the abemaciclib arms. The majority of neutropenia-related AEs were grade 1 or 2. In the abemaciclib + NSAI arm, only one patient experienced febrile neutropenia (grade 3). Due to neutropenia, four patients in the abemaciclib plus NSAI arm and none in the abemaciclib plus fulvestrant arm discontinued the study medication. Diarrhea of grade 3 was reported for 3.9% and 1.9% of patients in the abemaciclib arms. SAEs were observed in 19.5% and 15.4% of patients in the abemaciclib arms, respectively, and 9.1% and 7.5% in the placebo arms. The most often reported SAE was lung infection. Two fatalities in the abemaciclib plus NSAI arm (one with lung infection and one with dyspnea) and one death (lung infection) in the abemaciclib plus fulvestrant arm that occurred during study treatment or within 30 days of treatment ending were classified as treatment-related [48].

MonarchE is a randomized, open-label, phase III study of abemaciclib combined with standard adjuvant ET versus standard adjuvant ET alone in patients with high-risk, node-positive, early-stage HR+/HER2− BC [49]. In the abemaciclib arm, 463 patients (16.6%) stopped abemaciclib due to AEs. A total of 5141 patients (97.9% in the abemaciclib arm and 86.1% in the control arm) encountered at least one treatment-emergent AE. In the abemaciclib arm, the most common AEs were diarrhea, neutropenia, and fatigue. Grade ≥ 3 AEs occurred in 45.9% of abemaciclib arm participants and 12.9% of control arm patients. SAEs occurred in 12.3% of abemaciclib patients and 7.2% of control patients, with pneumonia being the most commonly reported SAE in both groups (0.8% and 0.5%, respectively). Deaths during trial treatment or within 30 days of termination were evenly distributed between the arms, with 14 (0.5%) occurring in each. The abemaciclib arm had 11 AEs (2, diarrhea and pneumonitis, were considered related to study medication) against 7 in the control arm [50].

As seen from all clinical studies, abemaciclib has been shown to increase serum creatinine. This is due to inhibition of tubular secretory transporters and does not affect glomerular function. In clinical studies, an increase in serum creatinine was observed during the first 28-day cycle of abemaciclib administration. Serum creatinine remained elevated but remained stable during treatment and was reversible when treatment was discontinued [51].

We present below (Table 3) the adverse reactions (≥10%) reported in the MONARCH-2, MONARCH-3, MONARCH PLUS, monarchE, and NEXT MONARCH studies in the abemaciclib arms.

## 5. Discussion

### 5.1. Lack of Predictive Biomarkers—A Solving Problem?

For patients with HR+/HER2 metastatic and high-risk early breast cancer (EBC), there are a number of problems that need to be addressed with the growing clinical usage of CDK4/6 inhibitors. Although many efforts have been made to pinpoint the mechanism of CDK4/6 inhibitor resistance, none of the potential mechanisms have been properly verified. Patients who are more likely to respond to treatment with CDK4/6 inhibitors may be found via the discovery of specific biomarkers. Finding the biomarkers for response prediction by tissue or liquid biopsy has been attempted on numerous occasions. However, no distinct biomarker has yet been found [52].

The demand for appropriate therapy for patients whose disease progresses while using CDK4/6 inhibitors has increased with the adoption of these drugs. It will be crucial to ascertain whether more CDK4/6 inhibitors might be used in future therapies. Future research and development should focus on selecting the right pharmaceuticals for use in conjunction with CDK4/6 inhibitors, as a number of novel medications are being tested in the clinic [51].

### 5.2. Where Does the Difference in Efficacy of CDK4/6is Come from?

While CDK6/cyclin D3 is linked to the hematopoietic system and is crucial for healthy thymus development and the maturation of bone marrow hematopoietic stem cells, CDK4/cyclin D1 is related to cell proliferation and is essential for supporting the progression of breast cancer. Abemaciclib has 14 times more potency against CDK4 than CDK6 and functions as a competitive inhibitor of both CDKs’ ATP-binding domains. This CDK4 selectivity suggests that abemaciclib may significantly reduce the growth of breast cancer cells [53,54]. Additionally, the considerably milder CDK6-inhibiting impact of abemaciclib might lessen the risk of bone marrow toxicity. These characteristics suggest that abemaciclib might be the first selective CDK4/6 inhibitor to allow continuous dosing to induce persistent target inhibition. When CDK4/6 is temporarily inhibited, the cell cycle can return to normal and DNA synthesis can resume. However, ongoing suppression of CDK4/6 will result in apoptosis and a persistent arrest of the cell cycle. In the cerebrospinal fluid and plasma, abemaciclib has similar drug concentrations and can also penetrate the blood–brain barrier [51]. Therefore, an important subject being investigated in clinical studies is whether abemaciclib may be useful in treating patients with brain metastases or in lowering the risk of getting brain metastases.

The efficacy results of several trials have been uneven in the metastatic scenario, despite the FDA’s approval of the CDK4/6 inhibitors palbociclib, ribociclib, and abemaciclib for the treatment of patients with metastatic HR-positive/HER2-negative breast cancer. The benefits of ribociclib, when combined with hormone therapy, were verified in the final analyses of the phase III MONALEESA-2 and MONALEESA-3 studies. The phase III PALOMA-2 trial of palbociclib plus letrozole, however, failed to achieve its OS end point in postmenopausal women with advanced breast cancer that was HR-positive/HER2-negative. Data from the phase III MONARCH 3 trial showed that data for abemaciclib plus letrozole or anastrozole trended toward a benefit in OS vs. letrozole or anastrozole alone, but the results did not reach statistical significance.

Regarding a potential explanation for these discrepancies, there is ongoing discussion about whether they can be attributable to a variable study design or to innate mechanistic variations. There are a number of plausible hypotheses that also recognize and take into consideration the risk involved in performing cross-trial comparisons. Disparities may be strongly influenced by inconsistent follow-up of the control group in PALOMA-2. Furthermore, variations in the PALOMA-2 patient’s illness-free period may indicate that individuals had earlier disease recurrence.

Palbociclib and ribociclib typically do not cause diarrhea toxicity, whereas abemaciclib does. Of course, the myelosuppression is lower. It is certainly a fantastic option for those who struggle with bone marrow. However, due to the daily dose frequency of abemaciclib, more aggressive disease may benefit from its treatment. Some subset analyses, such as those with palbociclib in the PALOMA-3 trial, suggest liver metastasis and a short disease-free interval in patients who haven’t had the most endocrine therapy-sensitive type of disease, and the real benefit in survival with those who have more endocrine therapy-sensitive disease. Thus, the answer to that question is still sort of forming [55].

### 5.3. Efficacy of CDK4/6is in the Visceral Crisis Setting

The ABC5 consensus states that the presence of visceral metastases alone is insufficient to establish the existence of a visceral crisis. Instead, critical organs must be seriously damaged before the demand for quick, reliable treatments arises. The following is how the ABC5 consensus describes a visceral crisis affecting the liver and lungs. When bilirubin levels rise significantly (>1.5 times the upper limit of normal) without Gilbert syndrome (also known as Meulengracht syndrome) or a biliary tract obstruction, a visceral crisis of the liver results. When dyspnea at rest worsens more quickly and cannot be eased by pleural drainage, a visceral lung crisis can be suspected [56].

Visceral crisis was not included in any of the clinical studies for breast cancer. Only a few case studies and retrospective research have been done on the management of visceral crises. One retrospective analysis, which was conducted in 2017, before any CDK4/6i products were approved by the FDA, came to the conclusion that chemotherapy had no appreciable effect on patient outcomes. Chemotherapy does not improve survival as compared to supportive care for HR+/HER2 metastatic breast cancer patients who are experiencing visceral crisis, according to another retrospective analysis [57].

The most reliable data for the treatment of visceral crises comes from a database that was retrospectively examined in real-world situations. A total of 336 individuals with visceral crisis were enrolled, and of those, 0.61 (18%) received CDK4/6 inhibitor therapy as initial treatment. Patients who received CDK 4/6i had a median OS of 11 months, compared to patients who did not, who had a median OS of 6 months. Patients who received CDK 4/6i treatment had a two-year OS of 26.1%, compared to 8.1% for chemotherapy patients. In comparison to chemotherapy, the use of CDK4/6is in patients with visceral crisis at diagnosis was associated with a 5-month improvement in OS [58].

Organ functions are affected, and performance status is compromised in patients with visceral crises. As a result, it is doubtful that these individuals would be able to withstand full doses of the most potent chemotherapy drugs, such as anthracycline and taxane. A retrospective examination of the real-world database indicates unequivocally that the combination of endocrine medicines and a CDK 4/6 inhibitor improves overall survival over chemotherapy. Finally, it is time to update the recommendations and take endocrine therapy with a CDK 4/6 combination into account as the best choice for the first management of the visceral crisis [59].

### 5.4. The Cost- Effectiveness Questions

The cost-effectiveness of CDK4/6is in the first-line management of HR+/HER2− metastatic BC in postmenopausal women in the US was examined in a research study by Masukar et al. in 2023. They calculated quality-adjusted life-years (QALYs) and incremental cost-effectiveness ratios using clinical efficacy and quality-of-life scores (utility) data from clinical trials, Medicare charges reported in US dollars per 2022 valuation, and a discount rate of 3% applied to costs and outcomes. Their model indicates that the combination of CDK4/6 inhibitors and letrozole is not cost-effective, with a modest increase in quality-adjusted life-years (QALYs) at a high cost, at a willingness to pay of $100,000/QALY gained [60].

### 5.5. SONIA Trial—A Changing Point of View?

The second line may be the optimal setting for patients with HR+/HER2− advanced breast cancer to receive CDK4/6is, according to findings from the phase III SONIA trial presented at the 2023 ASCO Annual Meeting [61]. These findings demonstrated that the frontline use of CDK4/6is prolongs the time on CDK4/6is and could increase costs and toxicities. Additionally, first-line treatment with CDK4/6 inhibition plus AI, followed by fulvestrant, did not improve PFS, OS, or quality-of-life (QOL) vs. second-line CDK4/6 inhibition plus fulvestrant after first-line AI therapy. Moreover, the safety profile was reported to be consistent for CDK4/6 inhibitors. The most common AEs were neutropenia, liver function abnormalities, anemia, and thrombocytopenia. The total number of grade 3 or higher AEs was 2782 when CDK4/6 inhibition was used in the frontline. Comparatively, there were 1620 grade 3 or higher AEs when this treatment was used in the second line. In total, there was a 42% decrease in the rate of grade 3 or higher AEs.

### 5.6. Future Perspectives

Numerous questions about CDK4/6 inhibitor resistance are still unresolved despite tremendous advances. It is still unknown how the various ET and targeted drug combinations compare to one another, to chemotherapy delivered with a single agent, or to the most promising CDK4/6 inhibitors. We fervently expect that CDK4/6 inhibitors will expand their application to additional cancer types in addition to HR+/HER2− breast cancer. Abemaciclib was recently licensed by the FDA as the first and only CDK4/6 inhibitor for use in HR+/HER2 node-positive early breast cancer patients with a Ki-67 score of less than 20% and a high risk of recurrence [62].

In vivo, abemaciclib inhibited tumor growth in multiple human cancer xenograft models, including those derived from non-small cell lung cancer (NSCLC), melanoma, glioblastoma, mantle cell lymphoma, and ER+ breast cancer [63].

Furthermore, it is not known whether abemaciclib’s biological capacity to penetrate the blood–brain barrier will one day make it a viable therapy choice for individuals with brain metastases due to breast cancer [51].

### 5.7. Drawbacks and Pitfalls of Clinical Trials

Although of crucial importance in the clinical outcome of real-world experience, these phase II and III clinical trials have their pitfalls and drawbacks. One of them is represented by the absence of including minority populations of different ethnicities; this could be the cause of yet unknown resistance mechanisms. Another drawback of the clinical trials in general is represented by the exclusion of pregnant women due to lack of insufficient in vitro data. One of the most evident and regretful drawbacks of clinical trials is the absence of head-to-head comparisons between the three FDA-approved CDK4/6is. In the absence of head-to-head comparisons, differentiating factors may include cross-study evaluation of efficacy as well as of AE profiles, dosing and administration considerations, drug interactions, and cost. Nonetheless, none of these clinical trials included patients with visceral crisis; therefore, no data could be extracted regarding the benefit of CDK4/6is in the visceral crisis setting.

### 5.8. Long Story Short Regarding Toxicities of CDK4/6is

For a better understanding of the frequency with which adverse reactions occur with the three CDK4/6 inhibitors, we performed statistics using Microsoft Excel v.2307 for each one separately.

Thus, for palbociclib, the five most frequent adverse events reported are neutropenia, leukopenia, fatigue, infections, and anemia (Figure 2).

For ribociclib, the five most frequent adverse events were neutropenia, nausea, leukopenia, fatigue, and diarrhea (Figure 3).

In the abemaciclib arms of clinical studies, the five most frequent adverse events were diarrhea, neutropenia, leukopenia, anemia, and fatigue (Figure 4).

For all three CDK4/6 inhibitors, the percentage of grade ≥ 3 adverse events is much lower than adverse events of any grade, which also include grades 1 and 2.

All three inhibitors have neutropenia and leukopenia as common side effects. Palbociclib and ribociclib have neutropenia as the most reported adverse reaction, while diarrhea is the most reported for abemaciclib. These findings are in line with the frequency of adverse events reported in the literature.

## 6. Conclusions

To the best of our knowledge, this is the latest and newest literature review compounding all phase II and III randomized clinical trials regarding the three CDK4/6 inhibitors’ toxicities. This review will hopefully be of real help to all clinicians in managing the adverse effects of CDK4/6 inhibitors by providing an easier way to research the reported percentage of specific adverse reactions in consecrated clinical trials.

CDK4/6 inhibitors are generally well tolerated. The most common AEs encountered on CDK4/6 inhibitor treatment include neutropenia, leukopenia, anemia, thrombocytopenia, fatigue, gastrointestinal side effects, hepatotoxicity, and QTc prolongation. However, there are some distinctions between palbociclib, ribociclib, and abemaciclib. The most prevalent adverse effect of palbociclib and ribociclib is neutropenia, while for ribociclib, it is QTc prolongation and hepatobiliary toxicity, and for abemaciclib, gastrointestinal toxicity. All these AEs can be very well managed due to the possibility of multiple dosage decreases and adjustments. Therefore, in clinical practice, acknowledging a side effect is of paramount importance to best manage it.

Moreover, future clinical trials should explore the use of CDK4/6is in the setting of visceral crisis, as they might increase OS in these patients, being a better solution. The cost-effectiveness balance is another issue that should be addressed, especially when speaking of CDK4/6is in the first-line setting. As we have seen from the SONIA trial, it challenges the need for first-line use of a CDK4/6 inhibitor, leaving us questioning whether we might change our point of view in the near future.

All in all, we believe that the paradigm of HR+/HER2− advanced breast cancer was completely changed for the better in the last decade with the introduction of CDK4/6is, and we are looking forward to seeing what the future brings for these inhibitors.

## Figures and Tables

**Figure 1 biomolecules-13-01422-f001:**
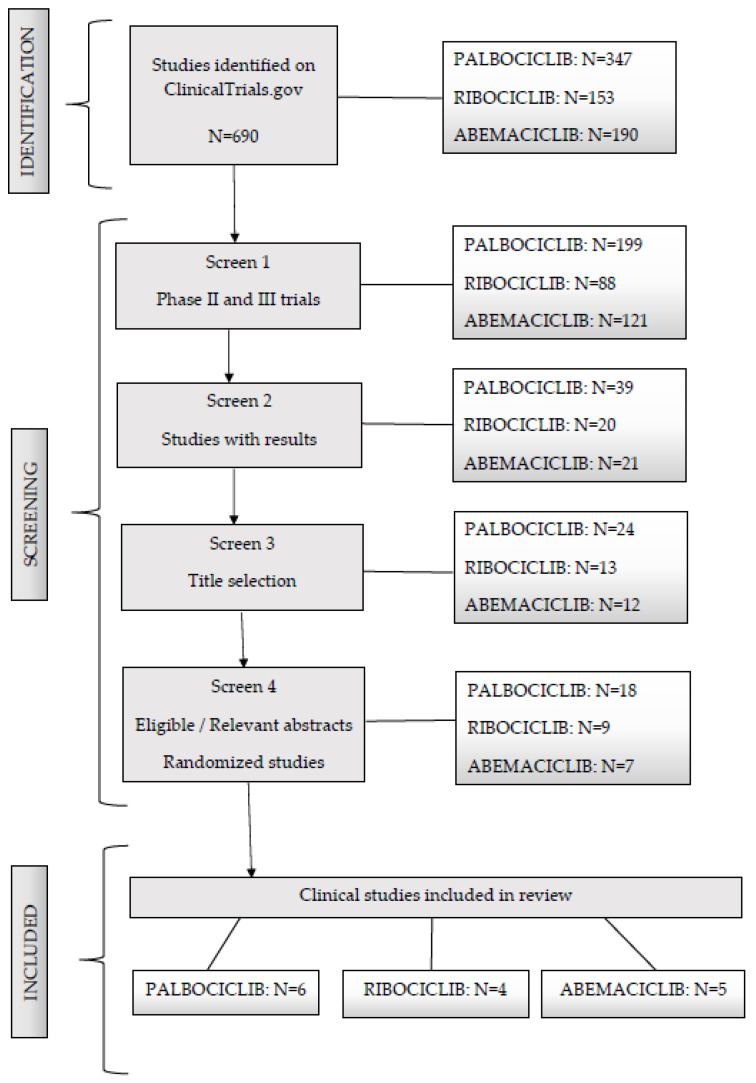
Consort flow diagram according to PRISMA 2009 guidelines.

**Figure 2 biomolecules-13-01422-f002:**
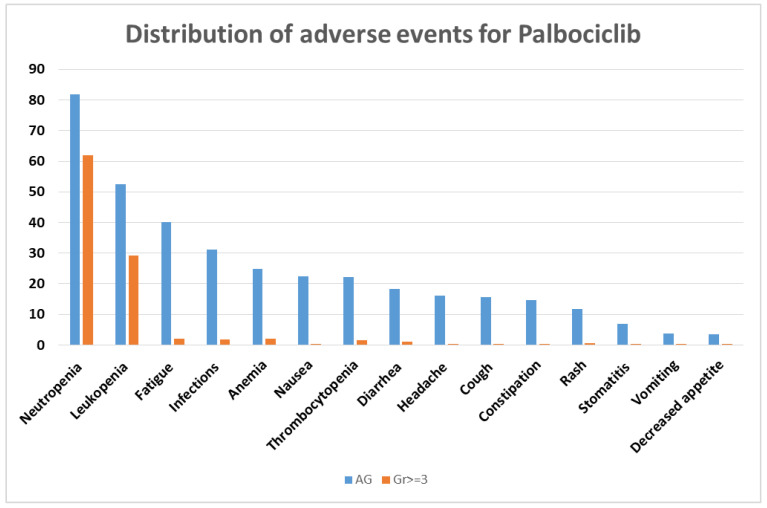
Distribution of adverse events for palbociclib. AG = any grade; Gr ≥ 3 = grade ≥ 3.

**Figure 3 biomolecules-13-01422-f003:**
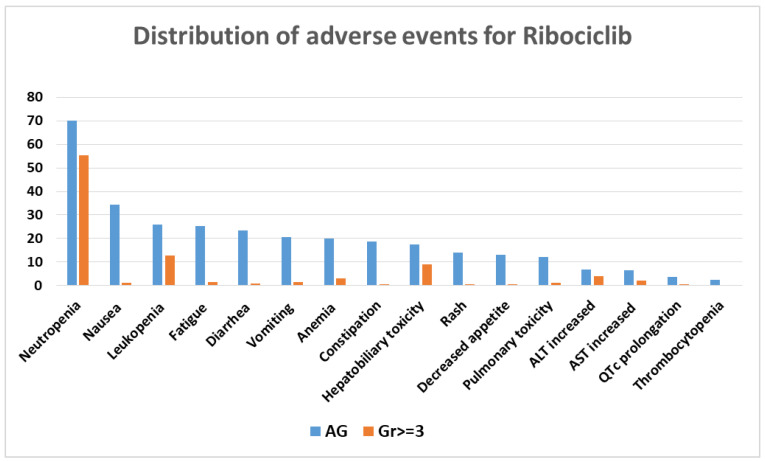
Distribution of adverse events for ribociclib. AG = any grade; Gr ≥ 3 = grade ≥ 3; AST = aspartate aminotransferase; ALT = alanine aminotransferase.

**Figure 4 biomolecules-13-01422-f004:**
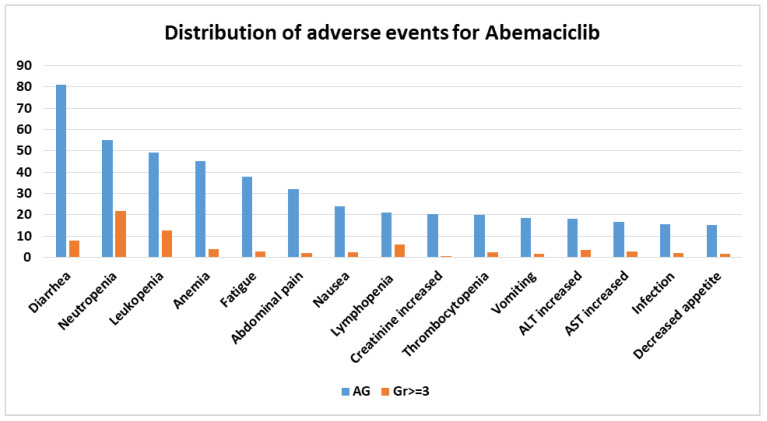
Distribution of adverse events for abemaciclib. AG = any grade; Gr ≥ 3 = grade ≥ 3; AST = aspartate aminotransferase; ALT = alanine aminotransferase.

**Table 1 biomolecules-13-01422-t001:** Adverse reactions (≥10%) reported in PALOMA-1, PALOMA-2, PALOMA-3, PALOMA-4, PALLAS, and PALLET studies for palbociclib arms. N = number of patients; % = percentage of patients; NA = not applicable; ET = endocrine therapy; AG = any grade; Gr 3 = Grade 3; Gr 4 = Grade 4.

	PALOMA-1			PALOMA-2			PALOMA-3			PALOMA-4			PALLAS			PALLET	
	Palbociclib + Letrozole N = 83			Palbociclib + Letrozole N = 444			Palbociclib + Fulvestrant N = 345			Palbociclib + Letrozole N = 169			Palbociclib + ET N = 2840			Palbociclib + Letrozole N = 201	
Adverse Events	AG %	Gr 3 %	Gr 4 %	AG %	Gr 3 %	Gr 4 %	AG %	Gr 3 %	Gr 4 %	AG %	Gr 3 %	Gr 4 %	AG %	Gr 3 %	Gr 4 %	AG %	Gr ≥ 3 %
Infections	NA	NA	NA	60	6	1	47	3	1	31	3	0	28	1	0	NA	NA
Neutropenia	75	53	6	80	56	10	83	55	11	98	72	13	83	57	4	55	41
Leukopenia	43	18	0	39	24	1	53	30	1	86	36	1	55	30	1	24	6
Anemia	35	5	1	24	5	<1	30	4	0	46	5	0	24	1	0	10	0
Thrombocytopenia	19	4	0	16	1	<1	23	2	1	49	5	2	22	1	0	15	0
Decreased appetite	21	1	0	15	1	0	16	1	0	NA	NA	NA	NA	NA	NA	NA	NA
Stomatitis	NA	NA	NA	30	1	0	28	1	0	16	0	0	NA	NA	NA	10	1
Nausea	30	2	0	35	<1	0	34	0	0	NA	NA	NA	20	0.5	0	25	0
Diarrhea	22	4	0	26	1	0	24	0	0	11	1	0	17	1	0	15	1
Constipation	16	1	0	20	0.5	0	22	0	0	NA	NA	NA	14	0	0	13	0
Vomiting	18	0	0	16	1	0	19	1	0	NA	NA	NA	NA	NA	NA	NA	NA
Rash	NA	NA	NA	18	1	0	17	1	0	NA	NA	NA	12	0.5	0	NA	NA
Fatigue	41	5	2	37	2	0	41	2	0	10	1	0	41	2	0	58	2
Headache	NA	NA	NA	22	0.2	0	29	1	0	NA	NA	NA	15	0.2	0	19	0
Cough	NA	NA	NA	25	0	0	23	0.3	0	16	1	0	14	0	0	11	0

**Table 2 biomolecules-13-01422-t002:** Adverse reactions (≥10%) reported in MONALEESA-2, MONALEESA-3, MONALEESA-7, and CORALLEEN studies for ribociclib arms. N = number of patients; % = percentage of patients; NA = not applicable; AST = aspartate aminotransferase; ALT = alanine aminotransferase.

	MONALEESA-2			MONALEESA-3			MONALEESA-7			CORALLEEN		
	Ribociclib + Letrozole N = 334			Ribociclib + Fulvestrant N = 483			Ribociclib + Endocrine Therapy N = 335			Ribociclib + Letrozole N = 51		
Adverse Events	Any Grade %	Grade 3 %	Grade 4 %	Any Grade %	Grade 3 %	Grade 4 %	Any Grade %	Grade 3 %	Grade 4 %	Any Grade %	Grade 3 %	Grade 4 %
Neutropenia	65.3	43.1	9.0	69.6	46.6	6.8	77.3	51.9	11.6	57	43	0
Nausea	55.1	2.7	0	45.3	1.4	0	NA	NA	NA	20	0	0
Fatigue	43.1	3	0.3	31.5	1.7	0	NA	NA	NA	14	0	0
Diarrhea	40.7	2.4	0	29	0.6	0	NA	NA	NA	14	0	0
Vomiting	35.0	3.9	0	26.7	1.4	0	NA	NA	NA	4	0	0
Constipation	29.9	1.2	0	24.8	0.8	0	NA	NA	NA	12	0	0
Anemia	24	3	0.6	17.2	3.1	0	22.4	3.6	0	8	0	0
Decreased appetite	22.2	1.5	0	16.1	0.2	0	NA	NA	NA	8	0	0
ALT increased	20.4	9.6	1.8	NA	NA	NA	NA	NA	NA	26	14	6
Rash	20.4	0.9	0	18.4	0.4	0	NA	NA	NA	24	2	0
AST increased	20.1	5.4	0.9	NA	NA	NA	NA	NA	NA	22	10	0
Leukopenia	17.1	9.3	0.6	28.4	13.5	0.6	34.9	14.9	1.2	NA	NA	NA
Thrombocytopenia	NA	NA	NA	NA	NA	NA	9.3	0.6	0.3	0	0	0
QTc prolongation	NA	NA	NA	NA	NA	NA	13	2	0	6	2	0
Pulmonary toxicity	16	3	0	NA	NA	NA	27	1.2	0.3	2	1	0
Hepatobiliary toxicity	NA	NA	NA	24	14	0	28	11	1	NA	NA	NA

**Table 3 biomolecules-13-01422-t003:** Adverse reactions (≥10%) reported in MONARCH-2, MONARCH-3, MONARCH PLUS, monarchE, and NEXT MONARCH studies in the abemaciclib arms. N = number of patients; % = percentage of patients; NA = not applicable; Abema = abemaciclib; Fulv = fulvestrant; AI = aromatase inhibitor; NSAI = non-steroidal aromatase inhibitor; ET = endocrine therapy; T = tamoxifen; Abema-150 = abemaciclib 150 mg; Abema-200 = abemaciclib 200 mg; AST = aspartate aminotransferase; ALT = alanine aminotransferase.

	MONARCH-2			MONARCH-3			MONARCH PLUS						monarchE			NEXT MONARCH					
	Abema + Fulv N = 441			Abema + AI N = 327			Abema + NSAI N = 205			Abema + Fulv N = 104			Abema + ET N = 2791			Abema + T N = 78		Abema-150 N = 79		Abema-200 N = 77	
Adverse Events	AG %	Gr 3 %	Gr 4 %	AG %	Gr 3 %	Gr 4 %	AG %	Gr 3 %	Gr 4 %	AG %	Gr 3 %	Gr 4 %	AG %	Gr 3 %	Gr 4 %	AG %	Gr ≥ 3 %	AG %	Gr ≥3 %	AG %	Gr ≥ 3 %
Diarrhea	86	13	0	81	9	0	80	4	0	80	2	0	82	8	0	54	1.3	67	4	62	9
Nausea	45	2.7	0	39	0.9	0	27	0	1	18	1	0	18	1	0	32	3	33	3	44	3
Abdominal pain	35	2.5	0	29	1.2	0	18	1	0	14	1	0	34	1	0	27	0	23	1	33	0
Vomiting	26	0.9	0	28	1.2	0	16	2	1	19	1	0	16	1	0	19	3	25	4	26	5
Infection	43	5	0.7	39	4	0.9	15	0	0	14	0	0	10	1	0	NA	NA	NA	NA	NA	NA
Fatigue	46	2.7	0	40	1.8	0	30	1	0	23	0	0	38	3	0	32	4	27	3	31	7
Decreased appetite	27	1.1	0	24	1.2	0	23	0	0	22	0	0	11	1	0	26	5	15	1	22	3
Creatinine increased	98	1.2	0	98	2.2	0	12	0	0	21	1	0	NA	NA	NA	18	1	11	0	10	0
Lymphopenia	63	12	0.2	53	7	0.6	17	6	0	21	11	1	13	5	0	NA	NA	NA	NA	NA	NA
Leukopenia	90	23	0.7	82	13	0	76	13	1	83	22	0	37	11	0	28	10	35	13	29	12
Neutropenia	87	29	3.5	80	19	2.9	80	26	1	81	29	1	45	18	1	42	23	54	30	52	38
Anemia	84	2.6	0	82	1.6	0	62	11	0	70	11	0	33	2	0	44	14	34	8	44	12
Thrombocytopenia	53	0.9	1.2	36	1.3	0.6	44	5	0	41	3	0	10	1	0	19	4	17	5	36	7
ALT increased	41	3.9	0.7	48	6	0.6	37	5	1	35	6	0	10	2	0	8	3	5	3	8	4
AST increased	37	3.9	0	37	3.8	0	37	4	1	31	2	1	10	2	0	10	3	5	0	10	3

## Data Availability

The data presented in this study are available in this article.

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
