# Peer review of "An Overview of the Safety Profile and Clinical Impact of CDK4/6 Inhibitors in Breast Cancer—A Systematic Review of Randomized Phase II and III Clinical Trials"

_biomolecules, 2023, doi:10.3390/biom13091422_

Round 1
Reviewer 1 Report
Dear Dr. Parosanu,
I have reviewed your submitted manuscrtpt. Please follow few of the suggestions.
Major point of concern is about the absence of any statistical correlation. Is this possible to include some statistical test hypothesis to make the study more meaningful. Minor are as follows:
1. Line 27…..1 in 39 (about 25%) chance that a woman will die of BC
It required re phrasing.
2. Line 32….will be diagnosed with invasive BC?
Please clear this sentence, if it is right.
3. General grammatical mistakes in using preposition e.g. Line 67…therapy might help over- 67 come the selectivity…could be with might help To…….
4. Line 479 need to remove instructions from the journal.
best regards,
Reviewer 2 Report
This review adresses an important subject, the toxicities of CDK4/6i. The authors have done a systematic search regarding studies that could contribute data for this review and they have collected the relevant data. Unfortunately the manuscript fails to translate this work into benefit for the reader. Here are the main reasons why this manuscript has to be rejected:
1. A section where the efficacy of the CDK4/6i is described is completely missing. There also differences in efficacy (and not their equivalence as stated in the introduction) could be highlighted.
2. The results section is an endless listing of toxicities, it would have been better to leave that to the tables that includebthe same information. Instead it would have been interesting to perform comparisons of selected toxicities in all three CDK4/6i.
3. Instead of a discussion the authors are presenting figures regarding the tox that is previously described in the text and also listed in the tables. In fact there is no discussion. The discussion would have been the section where the toxicities could have been put into the context of efficacy and efficacy differences. And it would have been the section where the treatment of acute visceral crisis could have been discussed (not chemotherapy as stated in the introduction), especially in the light of tox as this was supposed to be a review about side effects.
In summary unfortunately this is not a review but only a listing of side effects that are neither discussed nor put into context. The authors sholud consider writing a completely different manuscript and resubmit. I would be happy to review it then.
Reviewer 3 Report
Biomolecules (Manuscript ID: biomolecules-2587106), Comments to the Authors:
Title: An Overview of Safety Profile and Clinical Impact of CDK4/6 inhibitors in Breast Cancer - A Systematic Review of Randomized Phase II and III Clinical Trials
Comments
The submitted review discussed summary and update on the safety profile of the three CDK4/6 inhibitors, as it appears from all major phase II and III randomized clinical trials regarding palbociclib, ribociclib, and abemaciclib, including the most relevant 15 clinical trials
I think the submitted review can be accepted for publication after the authors respond to the following comments:
- The authors should indicate the rationale behind writing the review. The authors should explain why they initiated the idea of the review.
- The authors should provide a more in-depth analysis of their findings. They should highlight the most interesting results reported in certain trials in comparison to other results in other trials to give the reader a better understanding of the actual status of the drugs.
- The authors should provide their insights on the potential future application of the drugs in pharmaceutical field.
- The authors should highlight the drawbacks and pitfalls of the reported clinical trials.
- The authors should highlight their contribution to the research on the seeds.
- The authors should expand their discussion to inform analyze and criticize all previous trials results and provide a clear conclusion to the reader.
- There are some typos and grammatical errors that should be corrected.
- There are some typos and grammatical errors that should be corrected.
Round 2
Reviewer 1 Report
Dear Dr. Parosanu,
I reviewed your submitted review and I am satisfied that the responses of my suggestions are fully addressed.
Best regards,
Reviewer 2 Report
The authors have completely rewritten the manuscript and addressed all issues. I have no further comments.
Reviewer 3 Report
Biomolecules (Manuscript ID: biomolecules-2587106), Comments to the Authors:
Title: An Overview of Safety Profile and Clinical Impact of CDK4/6 inhibitors in Breast Cancer - A Systematic Review of Randomized Phase II and III Clinical Trials
Comments
After reading the authors' response to my comments, I think the revised review can be accepted for publication.